# Evolutionary Game Research on Green Construction Considering Consumers’ Preference under Government Supervision

**DOI:** 10.3390/ijerph192416743

**Published:** 2022-12-13

**Authors:** Xiaoxiao Geng, Ling Lv, Yingchen Wang, Ran Sun, Xiangmei Wang

**Affiliations:** 1School of Architecture and Art, Hebei University of Engineering, Handan 056038, China; 2School of Management Engineering and Business, Hebei University of Engineering, Handan 056038, China

**Keywords:** green construction, consumers’ green preference, government supervision, evolutionary game theory, construction industry

## Abstract

Construction is closely related to people’s lives and public activities. With regard to issues of energy conservation, emission reduction, and sustainable development put forth, the word “green” is increasingly used in the construction industry. Green construction is an inevitable requirement for the sustainable development of the construction industry. In addition, the government regulation mechanism is also the key to the establishment and development of green construction. Thus, on the basis of evolutionary game theory, this paper constructs an evolutionary game model of developers, general contractors, and the government. Then, the evolutionary stability strategies under different conditions are discussed. Lastly, the evolution mechanism in the game process of the green construction system and the influence of relevant parameters on the evolution path of the game model are analyzed through numerical simulation. The results show that (1) when consumer preferences change, the evolutionary stability strategies also change. Consumer preference plays a positive role in the establishment and development of green construction, but a certain increase in consumer preference will lead to free-rider behavior. (2) The government’s control mechanism can not only effectively mobilize the enthusiasm of enterprises to participate but also effectively inhibit the free-rider behavior of enterprises; however, it cannot completely prevent the occurrence of free-rider behavior. (3) On the one hand, the government can advocate green consumption to promote the improvement of consumers’ green preferences; on the other hand, enterprises actively carry out technological innovation and equipment advances to enhance the likelihood that consumers will embrace green preferences to achieve the optimal ESS. This study not only provides good guidance for developers and general contractors to make optimal strategic choices under different consumers’ green preferences, but also provides a reference for the government to formulate reasonable regulatory policies. At the same time, it promotes the development of green construction and further promotes environmental protection.

## 1. Introduction

The process of urbanization in China is accelerating, which will serve to promote the advancement of many areas in our country, especially the construction industry [1]. In 2019, the relevant state departments issued the “Green Construction Guidelines” [2], and in 2012, the Ministry of Finance issued the “Implementation Opinions on Accelerating the Development of Green Buildings in China”. At present, China has issued the latest “Opinions on Promoting Sustainable and Healthy Development of the Construction Industry” [3]. This puts green construction in a very important position from the system and greatly promotes the development of green construction in our country. However, due to the continuous expansion of the construction scale of construction industry, the demand for various resources is also growing. Affected by various factors during the construction process, it is difficult for China’s construction enterprises to effectively protect the environment during the construction process, which may lead to pollution of the surrounding environment [1,4]. This is contrary to China’s established development concepts and goals of energy conservation, emission reduction, and green environmental protection. Therefore, we should actively respond to the call for green buildings.

Green construction is an important link in the implementation process of construction projects, and the degree to which it is increasingly accepted allows for the results of green planning and design to come into practice. At the same time, it is also the most direct stage of environmental impact and resource and energy consumption, with the characteristics of large consumption, long cycle, more construction waste, and direct impact on the surrounding environment. According to research, in the whole life cycle, the energy consumption in the construction stage accounts for 23% of the whole life, while, in some buildings, energy consumption reaches just 40–60% [5]. Therefore, in the construction project, the green construction stage is an important link that directly affects the green degree of building products and the whole project. However, consumers are the end-users of construction products [6]. At the same time, on the demand side of construction products, the consumption industry is also the driving force for the construction and development of the green supply chain in the construction industry [7]. Therefore, the enterprise must change traditional construction methods and obtain a competitive advantage through the coordinated construction of a green construction system to better meet the needs of users. However, In the process of building green construction, due to the pursuit of maximum benefits, there has been a “free-ride” phenomenon, leading enterprises to choose to “enjoy the benefits” at little cost to them [8]. For example, to maximize its own material benefits, the company will carry out non-green construction because the other party carries out green construction during construction project implementation. Therefore, the government is required to regulate the main body of the green construction system to ensure the establishment and sustainability of green construction [9].

There are two kinds of government control measures: one is to reward companies that implement green construction, and the other is to punish companies that implement non-green construction [10]. Therefore, under the government’s reward and punishment mechanism, how should the construction unit and the construction unit respond? Who is actively participating in green construction? How will this affect the final equilibrium point? On this basis, it is regarded as a complex system composed of enterprises and government, and the abovementioned problems have become the key issues regarding whether green construction can be smoothly promoted. Therefore, it is meaningful to incorporate the decision-making and evolution process of developers, general contractors, and the government into the green construction system of the construction supply chain and conduct simulations [11,12].

This paper analyzes the impact on the equilibrium point of system evolution from the perspective of considering consumers’ green preferences. Under the premise that the participants show limited rationality, a three-party game model is established. The evolution equilibrium state of the complex system is obtained through theoretical analysis. The effects of different parameters on participants’ strategy evolution under different scenarios were studied. We put forward constructive suggestions to promote the establishment and development of green construction systems.

The remainder of the paper is structured as follows: Section 2 reviews the literature; Section 3 introduces the game relationship among developers, general contractors, and the government, as well as the model construction and theoretical analysis; Section 4 analyzes the stability of each equilibrium point and determines the evolutionary stability strategy in different scenarios; Section 5 presents the numerical simulation and discusses the influence of relevant parameters on the game’s evolutionary path; lastly, Section 6 summarizes the conclusions and limitations of this paper.

## 2. Literature Review

### 2.1. Green Construction

At present, the present research on green construction is mainly focused on cost-effectiveness, green construction technology, green construction management, green construction evaluation and other aspects, while there is less research on green construction supervision. In terms of cost-effectiveness, Kim Won Tae [13] defined and classified various costs of green construction. Through case studies and cost-benefit report studies in developed countries, the problems existing in South Korea’s green construction cost data were put forward, and corresponding suggestions were given. Qiao Qiang [14] proposed the concept of construction process cost and analyzed the composition of the cost. Xu Lei [15] conducted research on the countermeasures for developing green construction from the perspective of incremental cost. The research results showed that the factors that motivate incremental cost are composed of two levels, enterprise and government, and the government should formulate mandatory and incentive policies to promote the development of green construction. Li Pengjuan [16] analyzed the green construction measures in the “Evaluation Standard for Green Construction of Construction Projects”, divided the measures into incremental cost measures and non-incremental cost measures from the perspective of incremental cost, and proposed strengthening the management of incremental cost measures, which is beneficial in controlling the whole cost of green construction. Sang Peidong [17] used questionnaires to investigate people’s understanding of the importance of project manager skills. Structural equation modeling was selected to test the assumptions. The results showed that China is in the early stage of GBC development. The project manager’s leadership and organization, management by objectives, and EQ were considered important factors affecting green building performance. In terms of green construction technology, Wang Yujing [18] identified and listed 21 barriers in a questionnaire survey through a literature review and collected 225 effective responses from 21 provinces in China. The results of statistical analysis showed that the five major obstacles to the adoption of GCT were “lack of government incentives”, “additional costs associated with GCT”, “reliance on traditional construction technology”, “lack of technical training for project personnel”, and “conflicts of interest among stakeholders in the process of adoption of GCT”. Jong et al. [19] used a machine learning technique based on Bayesian inference to predict the optimal strength gain of sustainable geological materials and verified the wide application of this method in considering efficiency and sustainability. The final results showed that, although the geographical materials had very different mixed design requirements, the proposed Bayesian method was reliable and could accurately predict the strength of the geological materials. In terms of green construction management, Bon Gang Hwang [20] studied the knowledge and skills that project managers must have in green construction by means of literature research, investigation, and interviews with project managers. The research results showed that knowledge or skills such as schedule and management, stakeholder management, and communication management are very important. Xueying Wu [21] used a mixture of quantitative and qualitative research methods and Pearson correlation, regression analysis, and Student’s t-test analysis techniques to determine the relationship between green building management and highway engineering project quality. The study found that financial issues, design specifications and standards, and the impact of various risks are the most effective green performance building strategies. Pan Jun [22] studied the prevention and control of solid waste, noise, and air. In the aspect of solid waste, he used the idea of a circular economy for reference to realize reduction, reuse, and recycling. In terms of noise, construction noise should be controlled from three aspects: sound source, transmission route, and receiver. In terms of air prevention and control, prevention should be given priority. Zhang Tonghua [23] believed that attaching great importance to green construction management is beneficial to improving the market competitiveness of construction enterprises through the review of “Safety Technology of Construction Engineering and Green Construction”. In the evaluation of green construction, Kim [24] proposed the economic evaluation method of green construction based on LCC and LCA and considered that the economic value generated by improving environmental quality should be considered in the evaluation of the feasibility of green construction technology. Li Yan [25] introduced BIM cloud services, submodules, and overall architecture into the green building evaluation system. Through the transformation and integration of various computer protocol data storage methods, the results show that the cloud computing server can efficiently and accurately evaluate traditional green building evaluation methods. Setiawan [26] adopted the concept of the capability maturity model (CMM) and proposed the green construction capability model (GCCM) assessment to evaluate contractors’ ability to develop green buildings. Li Huiling [27] constructed an evaluation index system of green construction from the perspective of influencing factors of green construction, obtained the index weight through the gray clustering method, and constructed a cluster vector model to evaluate the green construction grade, and the results were consistent with the actual situation. On the driving force of green construction, Xiaer Xiahou [28] discussed the driving factors of construction industrialization in China by combining qualitative and quantitative methods, identified 15 factors, and divided them into three aspects. The results showed that the development of CI in China was driven by macro-development and government, and it was a self-driven process. Yongfu Li [29] analyzed the relationship between the 10 limiting factors, classified their driving forces through the ISM and MICMAC model, and finally divided them into six levels, among which the relevant part of the government was the surface level, and the weak professional ability was the deep level. Hawang [30] established structural equation model (SEM), ran the model with Amos software, and finally got the driving factors of green building development in Xinjiang and the driving degree of each driving factor. The research results revealed the following order: external driving force (all stakeholders > internal driving force (enterprise) > internal driving force (residents) > internal driving force (other stakeholders). Kamranfar [31] used decision-making trial and evaluation laboratory analysis, Delphi technology, and the ANP mixed decision-making method to analyze the building development obstacles under different economic, environmental, cultural, and social standards. The results showed that the main economic obstacles ranked first, followed by cultural and social obstacles, and the management obstacles ranked third. Asyikin Mahat Noor Aisyah [32] determined the key factors affecting the progress and cost performance of green construction projects (GCPs) through quantitative methods. The research results showed that the most important factor influencing the cost performance of green buildings is the high initial cost of green building development.

### 2.2. Research on the Impact of Consumers’ Green Preference on the Implementation of the Green Supply Chain

Consumers’ green preferences play an important role in promoting enterprises to implement green supply chains. Doonan [33] showed that, in the supply chain, consumer demand is the most important influencing factor in the formulation of the environmental performance of the green supply chain. Anton [34], through the analysis of survey data, showed that the most important driving force for the development of enterprises to establish an environmental management system is pressure from consumers. He Binbin [35] studied the strategic choice of green marketing in the supply chain from the perspective of decentralized decision making and vertical cooperation based on evolutionary game theory and the Stackelberg model. The results showed that the improvement of public awareness of green environmental protection can promote the adoption of a green marketing model in the supply chain. At the same time, the vertical cooperation of upstream and downstream in the supply chain can further improve the green marketing tendency of enterprises and enhance the performance level of the whole supply chain. Liu Mingwu [36], through the construction of a two-level multinational green supply chain model and a comparative analysis of the impact on green supply chain decision making from the three perspectives of tariffs, power structure, and consumer preferences, found that the imposition of tariffs will reduce the green degree of products. However, the increase in consumers’ green preferences could offset some of the negative effects of tariffs. Chen Kebing’s [37] research results showed that, when the green sensitivity of demand is low, retailers can obtain the highest profit in the Nash game between green manufacturers and traditional manufacturers. However, as the sensitivity of green demand increases, retailers will obtain the highest profit under the leadership of green manufacturers, while the supply chain system will also obtain more overall profit. Li Lin [38] constructed a supply chain green innovation differential game model in the context of centralized decision-making, decentralized decision making, and cost sharing, considering consumers’ green preferences and reference price effects. The results showed that the change in consumer preference characteristics is an important factor in motivating supply chain members to make green innovation efforts. Ling Yantao [39] studied the product green degree differentiation design and pricing strategy under the different willingness of consumers to pay. The research showed that when consumers’ willingness to pay is low, the green commodity market cannot exist; when the willingness to pay is higher than a certain threshold, the enterprise can produce a single kind of green product, and, when the WTP continues to increase, enterprises can produce vertically differentiated products in terms of greenness to meet the differentiated needs of consumers.

### 2.3. The Application of Evolutionary Game Theory in Government Supervision

Wang Dezheng [40] took the consumer feedback mechanism as the research background, constructed a four-party game model including manufacturers, e-commerce, government regulators, and consumers, and studied the selection of e-commerce product quality supervision strategies. The research results showed that, under the true evaluation of consumers, an effective government supervision and punishment mechanism and a consumer feedback mechanism can stabilize product quality and safety while increasing government penalties, which can make participants evolve to the optimal strategy to a certain extent. Xu Ai [41] used game theory to construct a tripartite game model of government, enterprises, and consumers and studied the relationship among the three and the game state. The research showed that governments can increase the enthusiasm of enterprises and consumers to participate in the green supply chain by setting appropriate subsidies to enterprises, subsidies to consumers, and penalties for enterprises not implementing green supply chain management. Using evolutionary game theory, Xu Jianzhong [42] analyzed the cooperative innovation behavior of Zheng Industry–University–Research’s new energy vehicles from the two aspects of market mechanisms and government supervision. The research showed that government subsidies are beneficial to drive enterprises to carry out cooperative innovation of new energy vehicles; reasonable tax rates and administrative penalties under government supervision are conducive to promoting the stability of cooperative innovation of new energy vehicles. Tang Huiling [43] applied evolutionary game theory to study the behavior game between government and enterprises on emission reduction in a green supply chain. The research showed that only through the concerted efforts of the government, enterprises, and the public can the goal of strict government supervision and independent emission reduction be achieved by enterprises. Rao Weizhen [44] studied the evolutionary process of government supervision decision making on the cooperative distribution strategy selection of enterprise alliances by establishing an evolutionary game model of government supervision and the cooperative low-carbon distribution of enterprise alliances. The research results showed that a larger initial government supervision proportion leads to faster strategy stabilization by the enterprise alliance. Using evolutionary game theory, Tengfei Shi [45] explored the willingness of elderly people to participate in regulation, the status of privacy protection of platform service providers, the degree of government regulation, and the key factors influencing the balance of the three-party game system. The research results showed that government rewards and punishments can effectively promote the optimization of the game system, thus improving the level of privacy protection of the intelligent pension platform. Hongyu Long et al. [46] used evolutionary game theory to study the impact of green development performance (GDP) and the government’s reward and punishment mechanism on the decision-making process of production recycling units. The research results showed that the government’s reward and punishment mechanism effectively standardizes the decision-making process of production recycling units. At the same time, the incentive effect of the subsidy probability on recycling units is more obvious, and the effect of the supervision probability on improving the motivation of production units to actively participate is more significant. According to evolutionary game theory, Wang Yuting [47] studied the strategic choice of quality behavior between upstream and downstream construction market players from the perspective of government supervision, taking into account the anti-risk efforts of the players. The results showed that the current quality supervision system does not have a stable and balanced evolution, which is related to the current low efficiency of quality supervision in China. If the government’s supervision and rewards and punishments are strengthened, the volatility of the system can be effectively suppressed. Liang Xi [12] applied evolutionary game theory to study the evolutionary stability strategy of the government and developers under the government’s dynamic reward and punishment policy from the perspective of the green building supply side. The research results showed that, when the government adopts the static subsidy and static tax policy, the game system does not have the evolutionary strategy for stability. When the government then adopts the combination of dynamic subsidy and static tax, static subsidy and dynamic tax, or dynamic subsidy and dynamic tax, the system has an evolutionary stability strategy. The policy combination of dynamic subsidy and static tax is better than other policies in promoting the development of green building. Using evolutionary game theory, Xu Fengwei [8] and others constructed an evolutionary game model of suppliers and manufacturers cooperating in green production and operation under the consideration of consumers’ green preferences to explore the influence of government regulatory mechanisms on the evolutionary stability of both parties’ strategic choices. The results showed that there are many states of equilibrium with the change in input cost and revenue caused by consumer preference. At this time, the implementation of the green supply chain can be promoted by the government’s incentive and punishment system. Wei [48] studied the stakeholders’ interest boundary and decision-making process by establishing a tripartite game model among government, developers, and consumers. The results show that the government’s policy cost cannot promote the stability of the system, and consumers’ recognition of the ecological value of green buildings is an important factor. Under the background of a low-carbon policy, Shi Qianqian [49] established a dynamic game model between the government and construction enterprises to study the evolution and stability strategies of the government and construction enterprises in different situations. The results showed that the government can formulate appropriate low-carbon policies according to the maturity of the market, and the mixed policy of subsidies and taxes can promote the active participation of enterprises.

To sum up, some researchers on green construction at home and abroad mainly focused on the cost-effectiveness of green construction, green construction technology, green construction management, green construction evaluation, green construction driving force, etc. From the existing research, it can be seen that the sustainable development of green construction industry and environmental protection are of great significance, which has been highly valued by many scholars. Moreover, scholars also put forward some macro-control strategies and improvement methods for the existing problems of green construction in China. These studies have played an important role in guiding the green construction and provided a good theoretical basis for us to continue the research on green construction. However, there are still some shortcomings in the current research. On the one hand, scholars’ research on the driving force of green construction seldom considered the incentive effect of consumers’ green preference on green construction. Judging from the law of economic development, consumers’ consumption demand is the main driving force to promote the reform of economic model. The green consumption demand of consumers promotes the implementation of green construction for resource conservation and environmental protection. On the other hand, the evolutionary game theory was applied to the relevant research of government supervision, and different game models were established to analyze the interest relationship between the players. In the existing research, most of the game models were established between the government and the general contractor or between the government and the core enterprise, which could not adequately reveal the supervision mechanism and had limited guiding effect on the supervision practice. In a word, the evolution of green construction was seldom discussed from the perspective of considering consumers’ green preference and government supervision. Therefore, from the perspective of consumers’ green preference as the driving force, this paper studies the stability and evolution mechanism of the strategic choices of developers, general contractors, and the government under different regulatory efforts by constructing a three-party evolutionary game model of green construction supervision, in order to provide theoretical support and practical guidance for the formation and development of green construction, and to further promote the healthy and sustainable development of the construction industry.

## 3. Model

Green building mainly refers to ensuring that the whole process of the project is a resource-saving and environment-friendly construction process under the guidance of the green concept, paying more attention to and pursuing the harmony between man and nature and between man and the environment. Therefore, during the whole construction phase, the general contractor should strictly carry out green construction in accordance with the concept of green environmental protection. On the premise of ensuring the engineering quality, green construction minimizes the negative impact of the whole process on the building itself and the surrounding environment by applying advanced construction technology. With “low energy consumption, low emissions, and low pollution” as the guiding principle, the coordination between the construction project and the environment was achieved through scientific and reasonable management and planning. Before construction, strict control should be carried out from the initial stages of material selection and planning, and the “three lows” should be achieved as much as possible in the whole construction process, so as to finally achieve the goal of green construction. During the implementation of green construction, the government plays a macro-control role. The government’s attention to green construction is an important factor in promoting the implementation of green construction.

The situation described in this paper is that developers, general contractors, and the government play games with each other to obtain expected revenues. The construction unit controls, supervises, and motivates the suppliers of materials and equipment and the general contractor through capital flow, information flow, and material flow after screening and determining the general contractor. The general contractor is especially engaged in the construction, demolition, and reconstruction of buildings [7]. To meet the demand for green buildings driven by consumers’ green preferences, both developers and general contractors are faced with the strategic choice of whether to pursue green construction. That is, whether the developers decide the green construction according to their own interests in the aspects of architectural design, raw material selection, material transportation, etc. on the original basis and whether the general contractor chooses green construction on the original production process and construction equipment according to their own interests. Green construction will increase the construction cost of the enterprise, which is contrary to the principle that the enterprise pursues the maximum benefit. Therefore, the government department needs to play a regulatory role and establish an incentive–punishment mechanism that can effectively promote the enterprise to carry out the green construction. That is, the government has two strategies; one is positive supervision, and the other is negative supervision, e.g., to reward and punish the behavior of both developers and general contractors, so as to promote both parties to carry out the green construction.

Every decision maker shows limited rationality, and they constantly adjust their strategies within the process of evolution. The model is an evolutionary game, as discussed in detail.

### 3.1. Hypotheses and Descriptions

The following assumptions are proposed to facilitate the evolutionary game model.

**Hypothesis** **1 (H1).**
*Consumers are rational and will adjust their preference for green products according to market changes.*


**Hypothesis** **2 (H2).**
*Hypothesis of strategy selection. The developers’ strategic choices are “green construction” and “non-green construction”; the general contractor’s strategic choices are “green construction” and “non-green construction”. The government has two possible strategies: one is to choose “active supervision”; the second is to choose “negative supervision”.*


**Hypothesis** **3 (H3).***If neither the developer nor the general contractor chooses the green construction strategy, the normal revenue of the developer and the general contractor is*$I_{1}$*and*$I_{2}$, respectively.

**Hypothesis** **4 (H4).**
*If both the developer and the general contractor choose the green strategy, the green preference of the consumer will pay a higher price for the green products so that both parties can obtain incremental revenue of*

$\theta I_{1}$

*and*

$\theta I_{2}$

*, respectively, and both parties will also invest more costs on the original basis, with cost increments of*

$C_{1}$

*and*

$C_{2}$

*, respectively. At the same time, regardless of whether the other party implements it or not, the implementing party will obtain the excess revenue*

E

*from the promotion of the corporate brand and the improvement of customer loyalty brought by the implementation of the green strategy.*


**Hypothesis** **5 (H5).***If only one party implements the green strategy, when the developer implements the green strategy, both parties can obtain the revenue increment caused by green preference of the consumers as*$\gamma \theta I_{1}$*and*$\gamma \theta I_{2}$*, respectively. When the general contractor implements the green strategy, both parties can obtain the (1 −*$\gamma )\;\theta I_{1}$*and (1 −*$\gamma )\;\theta I_{2}$*revenue increment caused by the consumers’ green preference, respectively. At the same time, the unimplemented party will obtain the spillover benefits brought by the green strategy of the implementer, i.e., benefit*D brought by the free ride of the unimplemented party due to the externality, including when the general contractor carries out the green construction the spillover benefits of the developer $D_{1}$*, and the excess benefits of the general contractor are*
$E_{4}$*. If only the developer carries out green construction, the general contractor’s overflow revenue is*
$D_{2}$*, and the developer’s excess revenue is*
$E_{3}$*; for developers,*
$D_{1}\;>\;E_{1}-E_{3}$*, and, for general contractors,*
$D_{2}\;>\;E_{2}-E_{4}$.

**Hypothesis** **6 (H6).**
*Under government supervision, certain subsidies*

$S_{1}$

*and*

$S_{2}$

*are given to developers and general contractors who implement green construction, and certain punishments*

$F_{1}$

*and*

$F_{2}$

*are given to developers and general contractors who implement non-green construction.*


**Hypothesis** **7 (H7).***When the government chooses to supervise, it will pay a certain supervision cost*$C_{3}$*. Regardless of the government**’s strategy, if developers or general contractors implement non-green construction, the government will have to bear the corresponding environmental pollution control costs*$C_{4}$*and*$C_{5}$.

The meanings of the parameters are shown in Table 1.

### 3.2. Construction of the Model

Under the condition of the bounded rationality assumption, the probability of developers choosing to implement the green strategy is α, and the probability of choosing to implement non-green strategy is 1− α. The probability that the general contractor chooses to implement green construction is β, and the probability that they choose to implement non-green construction is 1−β. The probability of the government choosing supervision is γ, and the probability of choosing formal policies is 1− γ. On the basis of these assumptions, the revenue matrix is shown in Table 2.

## 4. Evolutionary Game Model Analysis

### 4.1. Calculation of Stable Points

#### Stability Analysis of Developer

When a developer performs green construction, its expected revenue can be expressed as
(1)$$U_{1T}=\gamma S_{1}+I_{1}+x\theta I_{1}+E_{3}-C_{1}+\beta \left[\left(1-x\right)\theta I_{1}+E_{1}-E_{3}\right].$$

When a developer performs non-green construction, its expected revenue can be expressed as
(2)$$U_{1F}=-\gamma F_{1}+\beta \left(1-x\right)\theta I_{1}+\beta D_{1}+I_{1}.$$

The average revenue of the developer is
(3)$$\bar{U_{1}}=\alpha U_{1T}+(1-\;\alpha )U_{1F}=\alpha [\gamma \left(S_{1}+F_{1}\right)+x\theta I_{1}+E_{3}-C_{1}+\beta \left(E_{1}-E_{3}-D_{1}\right)]+[I_{1}-\gamma F_{1}+\beta \left(1-x\right)\theta I_{1}+\beta D_{1}].$$

According to the Malthusain principle, the developer’s replication dynamic equation is obtained as follows:(4)$$F(\alpha )\;=\frac{d\alpha }{dt}=\alpha (U_{1T}-\bar{U_{1}})=\alpha (1-\alpha )[\gamma \left(S_{1}+F_{1}\right)+x\theta I_{1}+E_{3}-C_{1}+\beta \left(E_{1}-E_{3}-D_{1}\right)].$$

When $\beta =\frac{x\theta I_{1}+E_{3}-C_{1}+\;\gamma \left(S_{1}+F_{1}\right)}{-\left(E_{1}-E_{3}-D_{1}\right)}$, F(α) =0, regardless of the value of α, both of the developers’ strategies are ESS [50,51]. When $\beta \;>\;\frac{x\theta I_{1}+E_{3}-C_{1}+\;\gamma \left(S_{1}+F_{1}\right)}{-\left(E_{1}-E_{3}-D_{1}\right)}$, let F(α) =0, and then α=0 and α=1 will be obtained; F(0) > 0, F(1) < 0, α=1 is the stability point. That is, developers’ green production is the equilibrium point. When $\beta \;<\;\frac{x\theta I_{1}+E_{3}-C_{1}+\;\gamma \left(S_{1}+F_{1}\right)}{-\left(E_{1}-E_{3}-D_{1}\right)}$, let F(α) =0, and then α=0 and α=1 will be obtained; F(0) < 0, F(1) > 0, α=0 is the stability point. That is, non-green production of developers is the equilibrium point.

Similarly, the general contractor’s replicated dynamic equations are available:(5)$$F(\beta )=\frac{d\beta }{dt}=\beta (U_{2T}-\bar{U_{2}})=\beta (1-\beta )[\gamma \left(S_{2}+F_{2}\right)+\alpha (E_{2}-E_{4}-D_{2})+\left(1-x\right)\theta I_{2}+E_{4}-C_{2}].$$

When $\alpha \;=\frac{\left(1-x\right)\theta I_{2}+E_{4}-C_{2}+\gamma \left(S_{2}+F_{2}\right)}{-(E_{2}-E_{4}-D_{2})},\;F(\beta )\;=0$, regardless of the value of β, both of the developers‘ strategies are ESS. When $\alpha >\frac{\left(1-x\right)\theta I_{2}+E_{4}-C_{2}+\gamma \left(S_{2}+F_{2}\right)}{-(E_{2}-E_{4}-D_{2})}$, let F(β) =0, and then β=0 and β=1 will be obtained; F(0) > 0, F(1) < 0, β=1 is the stability point. That is, developers’ green production is the equilibrium point. When $\alpha \;<\;\frac{\left(1-x\right)\theta I_{2}+E_{4}-C_{2}+\gamma \left(S_{2}+F_{2}\right)}{-(E_{2}-E_{4}-D_{2})}$, let F(β) =0, and then β=0 and β=1 will be obtained; F(0) < 0, F(1) > 0, β=0 is the stability point. That is, non-green production of developers is the equilibrium point.

The government’s replication dynamic equation is
(6)$$F(\gamma )=\frac{d\gamma }{dt}=\gamma (U_{3T}-\bar{U_{3}})=\gamma (1-\gamma )[\alpha \left(-S_{1}-F_{1})+\beta (-S_{2}-F_{2})-C_{3}+F_{1}+F_{2}\right].$$

When $\alpha \;=\frac{\beta (-S_{2}-F_{2})-C_{3}+F_{1}+F_{2}}{S_{1}+F_{1}}$, *F*(*γ*) = 0, regardless of the value of α, both government strategies are ESS. When $\alpha >\frac{\beta (-S_{2}-F_{2})-C_{3}+F_{1}+F_{2}}{S_{1}+F_{1}}$, F(γ)=0, γ=0 and γ=1 are obtained; F(0) > 0, F(1)<0, γ=1 is the equilibrium point, i.e., the government chooses active supervision as the equilibrium point. When $\alpha \;>\;\frac{\beta (-S_{2}-F_{2})-C_{3}+F_{1}+F_{2}}{S_{1}+F_{1}}$, F(γ)=0, γ=0, and γ=0 are obtained; F(0) > 0, F(1)<0, γ=1 is the stability point, i.e., The government chooses negative supervision as the equilibrium point. Similarly, when $\beta \;=\frac{\alpha (-S_{1}-F_{1})-C_{3}+F_{1}+F_{2}}{S_{2}+F_{2}}$, F(γ)=0, regardless of the value of α, both government strategies are ESS. When $\beta >\frac{\alpha (-S_{1}-F_{1})-C_{3}+F_{1}+F_{2}}{S_{2}+F_{2}}$, F(γ)=0, γ=0, and γ=1 are obtained; F(0) > 0, F(1)<0, and γ=1 is the stability point, i.e., the government chooses active supervision as the equilibrium point. When $\beta <\frac{\alpha (-S_{1}-F_{1})-C_{3}+F_{1}+F_{2}}{S_{2}+F_{2}}$, F(γ)=0, γ=0, and γ=1 are obtained; F(0)<0, F(1)>0, γ=0 is the stability point, i.e., the government chooses negative supervision as the equilibrium point.

### 4.2. Evolutionary Equilibrium Stability Analysis

According to the analysis method proposed by Friedman, the stability of the equilibrium point of the game can be determined by the local stability of the Jacobian matrix [52]. The partial derivatives of α, β, and γ in the replicated dynamic equation are calculated, and the Jacobian matrix J of the system is obtained as follows:(7)$$J=\left[\begin{array}{cc} \begin{array}{c} \frac{\partial F\left(\alpha \right)}{\alpha }\\ \frac{\partial F\left(\beta \right)}{\alpha } \end{array} & \begin{array}{cc} \begin{array}{c} \frac{\partial F\left(\alpha \right)}{\beta }\\ \frac{\partial F\left(\beta \right)}{\beta } \end{array} & \begin{array}{c} \frac{\partial F\left(\alpha \right)}{\gamma }\\ \frac{\partial F\left(\beta \right)}{\gamma } \end{array} \end{array}\\ \frac{\partial F\left(\gamma \right)}{\alpha } & \begin{array}{cc} \frac{\partial F\left(\gamma \right)}{\beta } & \frac{\partial F\left(\gamma \right)}{\gamma } \end{array} \end{array}\right].$$

According to the Jacobian matrix, if F(α)=0, F(β)=0, and F(γ)=0, eight pure strategy equilibrium solutions can be obtained: A1 = (0, 0, 0), A2 = (1, 0, 0), A3 = (0, 1, 0), A4 = (0, 0, 1), A5 = (1, 0, 1), A6 = (0, 1, 1), A7 = (1, 1, 0), and A8 = (1, 1, 1), along with a mixed strategy equilibrium solution A9 = (*α**, *β**, *γ**). According to the Lyapunov stability criterion, when the eigenvalue of the Jacobian matrix is nonpositive, the equilibrium point is the evolutionary stability point. For example, when A1 = (0, 0, 0), the Jacobian matrix can be derived as follows:(8)$$J_{1}=\left[\begin{array}{cc} \begin{array}{c} \lambda_{1} \end{array} & \begin{array}{cc} \begin{array}{c} \lambda_{2} \end{array} & \end{array}\\ & \begin{array}{cc} & \lambda_{3} \end{array} \end{array}\right],$$
where $\lambda_{1}=\left(1-2\alpha \right)[\gamma \left(S_{1}+F_{1}\right)+x\theta I_{1}+E_{3}-C_{1}+\beta \left(E_{1}-E_{3}-D_{1}\right)]$, $\lambda_{2}=\;(1\;-2\beta )[\gamma \left(S_{2}+F_{2}\right)+\;\alpha (E_{2}-E_{4}-D_{2})+\left(1-x\right)\theta I_{2}+E_{4}-C_{2}],$ and
$\lambda_{3}=\left(-2\gamma +1\right)[\alpha (-S_{1}-F_{1})+\beta (-S_{2}-F_{2})\;-C_{3}+F_{1}+F_{2}]$.

According to the equilibrium point, the available eigenvalues are shown in Table 3.

Combined with the size assumption of model parameters,$\;F_{1}+F_{2}>C_{3},\;S_{1}+C_{3}<F_{2},\;S_{2}+C_{3}<F_{1}$. Therefore, A1 = (0, 0, 0), A2 = (1, 0, 0), A3 = (0, 1, 0), and A8 = (1, 1, 1) are not asymptotically stable points.

In the process of building a green construction system, both developers and general contractors actively participate in it, which is beneficial to improve the green degree of green buildings and reduce the waste production in the construction production process to a certain extent. This is of great significance to the sustainable development of the environment. When consumers’ preference is low, because of the existence of “free-riding” behavior, and when one party carries out green construction and brings more excess benefits to the other party than the green construction itself, the enterprise usually chooses non-green construction strategy. In contrast, both parties will choose a green construction strategy. When at least one party carries out a non-green construction strategy, the government will pay a certain amount of environmental management fees. To reduce expenditures on governance costs, the government will choose an active regulatory strategy, but with the improvement of consumer preferences, both developers and general contractors will actively implement the green construction strategy, and the government will also adopt a passive regulatory strategy. Therefore, we can choose A7 = (1, 1, 0) as the optimal asymptotic stability point, i.e., (green construction, green construction, negative supervision) is the optimal ESS, and, only when all three eigenvalues are non-integers, A7 = (1, 1, 0) is the ESS. That is, when $\theta >\frac{C_{1}-E_{1}+D_{1}}{xI_{1}}$ and $\theta >\frac{C_{2}-E_{2}+D_{2}}{I_{2}-xI_{2}}$, the green construction system constructed by the three parties is in an optimal stable and balanced state.

(1)Stability analysis of A4 = (0, 0, 1)

When $0<\theta <\frac{C_{1}-E_{3}-S_{1}-F_{1}}{xI_{1}},\;0<\theta <\;\frac{C_{2}-E_{4}-S_{2}-F_{2}}{I_{2}-xI_{2}},F_{1}+F_{2}>C_{3}$, i.e., when the degree of consumers’ green preference is extremely low, A1 = (0, 0, 1) is the system evolution stability point (ESS). At this time, due to the lack of demand for green products due to the low green preference of consumers, the sum of the additional benefits and excess benefits brought by the green preference of consumers and the government subsidies is less than the incremental costs paid by both parties to implement the green construction strategy, i.e., the benefits obtained by developers or general contractors in implementing the green decision are too low or even lower than their respective input costs. In this case, neither party will carry out green construction. The analysis of $F_{1}+F_{2}>C_{3}$ shows that, if the cost of supervision is less than the fines from developers and general contractors, the government will choose an active supervision strategy.

(2)Stability analysis of A5 = (1, 0, 1)

When $\frac{C_{1}-E_{3}-S_{1}-F_{1}}{xI_{1}}<\theta <\frac{C_{1}-E_{1}+D_{1}-S_{1}-F_{1}}{xI_{1}},\;0<\theta <\frac{C_{2}-E_{4}-S_{2}-F_{2}}{I_{2}-xI_{2}}$, i.e., when the green preference of consumers has increased compared with the first case, where A6 = (1, 0, 1) is the ESS, the sum of the developer’s additional benefits and excess benefits brought by the green preference of consumers and government subsidies is greater than the incremental costs paid for implementing the green construction strategy, but the sum of all the developers’ benefits is less than the free-rider effect. The general contractor’s revenue situation is still the same as the first situation; hence, the developer chooses the green construction strategy, and the general contractor chooses the non-green construction strategy.

① If the government’s supervision of developers is strengthened and the supervision of general contractors is kept as is, i.e., when $S_{1}$ or $F_{1}$ is increased, the stable equilibrium point of the system will always be (1, 0, 1).

② If the government’s supervision over the general contractor is strengthened and the supervision over the developer is maintained as is, i.e., when $S_{2}$ or $F_{2}$ is increased to a certain extent, such that $\frac{C_{2}-E_{4}-S_{2}-F_{2}}{I_{2}-xI_{2}}<\theta <\frac{C_{2}-E_{2}+D_{2}-S_{2}-F_{2}}{I_{2}-xI_{2}}$, there is a stable point, but the stable point is not unique, i.e., (1, 0, 1) or (0, 1, 1). At this time, both parties may choose opportunistic behavior, resulting in one party choosing green construction and the other party “hitchhiking”. At this time, the final evolution result of the system is determined by the position of the saddle point and the initial point of the system. As shown in Figure 1, if the initial strategy selection falls within the area of M, the system eventually converges to (0, 1, 1). The developer selects the non-green construction strategy, and the general contractor selects the green construction strategy. If the initial strategy choice falls within the region of N, the system converges to (1, 0, 1), i.e., the developer chooses the green construction strategy, and the general contractor chooses the non-green construction strategy.
(9)$$S_{M}=\frac{1}{2}\left(\frac{C_{2}-E_{4}-\theta I_{2}+x\theta I_{2}-S_{2}-F_{2}}{E_{2}-E_{4}-D_{2}}+\frac{E_{1}-D_{1}-C_{1}+x\theta I_{1}+S_{1}+F_{1}}{E_{1}-E_{3}-D_{1}}\right).$$

According to Equation (9), the size of the area is affected by *x*, *θ*, *I*_1_, *I*_2_, *C*_1_, *C*_2_, *E*_1_, *E*_2_, *E*_3_, *E*_4_, *D*_1_, *D*_2_, *S*_1_, *S*_2_, *F*_1_ and *F*_2_. The correlation of the relevant parameters was judged by partial derivatives. There were 14 factors influencing the area of area *M*, among which one factor *θ* was uncertain, and the other 13 parameters had a monotonically increasing or decreasing relationship with the area of area *M*. The specific effects of these 14 parameters on the strategic choices of developers and general contractors are shown in Table 4.

As shown in Table 4, the sizes of parameters *x*, *I*_1_, *I*_2_, *C*_1_, *C*_2_, *E*_1_, *E*_2_, *E*_3_, *E*_4_, *D*_1_, *D*_2_, *S*_1_, *S*_2_, *F*_1_ and *F*_2_ affect the change of the saddle point and the area of region M. With the increase in parameters *I*_2_, *C*_1_, *E*_2_, *E*_4_, *D*_1_, *S*_2_, and *F*_2_, the saddle point moves leftward and upward, and the area of M increases. At this time, the probability that the system will converge to (0, 1, 1) is higher. With the decrease in parameters *x*, *I*_1_, *C*_2_, *E*_1_, *E*_3_, *D*_2_, *S*_1_, and *F*_1_, the saddle point moves leftward and upward, and the area of area M increases. At this time, the probability that the system will converge to (0, 1, 1) is higher, i.e., the developer is more willing to implement the non-green construction strategy, and the general contractor is more willing to implement the green construction strategy. In contrast, the greater the probability that the system converges to (1, 1, 0), developers are more willing to implement green construction strategies, and general contractors are more willing to implement non-green construction strategies.

(3)Stability analysis of A6 = (0, 1, 1)

When $0<\theta <\frac{C_{1}-E_{3}-S_{1}-F_{1}}{xI_{1}},\;\frac{C_{2}-E_{4}-S_{2}-F_{2}}{I_{2}-xI_{2}}<\theta <\frac{C_{2}-E_{2}+D_{2}-S_{2}-F_{2}}{I_{2}-xI_{2}}$, the consumer’s green preference increases compared with the first case, where A6 = (0, 1, 1) is the ESS. The sum of the additional revenue and excess revenue brought by the general contractor’s green preference of consumers and the government subsidy is greater than the incremental cost paid for implementing the green construction strategy, but the developer’s is still the same as the first case; thus, the general contractor chooses the green construction strategy, and the developer chooses the non-green construction strategy.

① If the government’s supervision over the general contractor is strengthened, and the supervision over the developers is kept as it is, i.e., when *S*_2_ or *F*_2_ is increased, the stable equilibrium point of the system will always be kept at (0, 1, 1).

② If the government’s supervision over the general contractor is strengthened and supervision over the developer is maintained as it is, i.e., when $S_{2}$ or $F_{2}$ is increased to a certain extent, such that $\frac{C_{2}-E_{4}-S_{2}-F_{2}}{I_{2}-xI_{2}}<\theta <\frac{C_{2}-E_{2}+D_{2}-S_{2}-F_{2}}{I_{2}-xI_{2}}$, there is a stable point, but the stable point is not unique, i.e., (0, 1, 1) or (1, 0, 1), which was analyzed above and is not analyzed here When *S*_2_ or *F*_2_ continues to increase, such that $\theta >\frac{C_{2}-E_{2}+D_{2}-S_{2}-F_{2}}{I_{2}-xI_{2}}$, the equilibrium point of the system will be stable at (1, 0, 1), i.e., the developer chooses the green construction strategy, and the general contractor chooses the non-green construction strategy. At this time, regardless of the strategy implemented by the other party, the developer has the highest benefit in choosing the green construction strategy, i.e., the sum of the profit and the government subsidy from green construction alone is greater than the difference between the free-rider benefit and the government penalty from choosing the non-green construction strategy. However, at this time, for the general contractor, although the sum of the profit and government subsidy brought by green construction alone is greater than the government penalty for choosing the non-green construction strategy but less than the difference between the free-rider benefit and government penalty, the general contractor will choose the non-green construction strategy.

## 5. Simulation Analysis and System Optimization

### 5.1. Stability Analysis of the Equilibrium Point

MATLAB is used to simulate the evolutionary game process of the three-party system, and numerical simulations are performed on A4 = (0, 0, 1), A5 = (1, 0, 1), and A6 = (0, 1, 1) to more clearly represent the evolutionary game behavior of the three-party system and verify the correctness of the game model.

#### 5.1.1. Numerical Simulation Analysis of A4 (0, 0, 1)

According to the asymptotic stability requirement of a4 = (0, 0, 1), the green construction system is simulated in combination with the actual situation, where *x* = 0.5, *θ* = 0.4, *I*_1_ = 10, *I*_2_ = 8, *C*_1_ = 5, *C*_2_ = 4, *E*_1_ = 3, *E*_2_ = 3, *E*_3_ = 1.8, *E*_4_ = 1.3, *D*_1_ = 2.5, *D*_2_ = 2.4, *S*_1_ = 0.01, *S*_2_ = 0.01, *F*_1_ = 0.65 and *F*_2_ = 0.65. The results obtained by the simulation software MATLAB (2019b) are shown in Figure 2.

The initial dynamic evolution trend is stable at the system equilibrium point A4 = (0, 0, 1) (as shown in Figure 2), indicating that ESS is a non-green construction decision, non-green construction, and active supervision. To make the balance closer to the optimal strategic goal, the government should strengthen the supervision of relevant enterprises and strengthen supervision. The government can encourage developers or general contractors to invest in green construction by increasing penalties or subsidies. On the basis of the stability analysis of A4 = (0, 0, 1), it can be seen from Figure 3a that, with the increase in penalty parameter *F*_1_ for developers, the three-party system converges to (1, 0, 1), indicating that the general contractor’s strategic choice may not be affected when the government increases its supervision over developers, and only developers in this system will actively participate in green construction. The simulation results in Figure 4a show that ESS has changed from the original (non-green construction, non-green construction, and active supervision) to (green construction, non-green construction, active supervision). As seen from Figure 3b, with the increase in the penalty parameter *F*_2_ for the general contractor, the three-party system converges to (0, 1, 1), indicating that, when the government increases the supervision over the general contractor, the developer’s strategic choice does not change, and only the developer will actively participate in green construction in this system. The simulation results shown in Figure 4b show that ESS changed from the original (non-green construction, non-green construction, and active supervision) to (non-green construction, green construction and active supervision), thus realizing the optimization of the game system.

#### 5.1.2. Numerical Simulation Analysis of A5 (1, 0, 1)

According to the asymptotic stability requirement of A5 = (1, 0, 1), the green construction system is simulated in combination with the actual situation, where *x* = 0.5, *θ* = 0.4, *I*_1_ = 10, *I*_2_ = 8, *C*_1_ = 5, *C*_2_ = 4, *E*_1_ = 3, *E*_2_ = 3, *E*_3_ = 1.8, *E*_4_ = 1.3, *D*_1_ = 2.5, *D*_2_ = 2.4, *S*_1_ = 0.01, *S*_2_ = 0.01, *F*_1_ = 1.5 and *F*_2_ = 0.65. The results obtained by the simulation software are shown in Figure 5.

The initial dynamic evolution trend is stable at the system equilibrium point A5 = (1, 0, 1) (as shown in Figure 5), indicating (green construction, non-green construction, active supervision) ESS. Through the above analysis, when the government continues to increase its supervision, it will increase the government’s supervision of developers and the government’s supervision of general contractors. On the basis of the stability analysis of A5 = (1, 0, 1), Figure 6a shows that, when the government’s penalty parameter *F*_1_ is increased for developers, developers will tend to choose a green construction strategy more quickly. Although the general contractor will slow down the speed of the non-green construction strategy, it will still stabilize in the non-green construction strategy in the end. The simulation results show that the ESS is still (green construction, non-green construction, active supervision). As seen from Figure 6b, when the government’s penalty parameter *F*_2_ for the general contractor is increased, both parties may tend to the green construction strategy at first with the increase in the parameter *F*_2_; however, due to the existence of free riding, when one party chooses the green construction strategy, the other party will tend to choose the non-green construction strategy. When the parameter *F*_2_ continues to increase and the effect of free riding by the general contractor is less than the government penalty, the general contractor will choose the green construction strategy no matter how the developer chooses the strategy. However, due to the free-rider benefit of developers, when the general contractor chooses the green construction strategy, it will choose the non-green construction strategy. At this time, the green construction system ESS will change from (green construction, non-green construction, and active supervision) to (non-green construction, green construction, and active supervision). From the analysis of the above results, the optimization of the system cannot be achieved by continuously increasing government supervision.

However, combined with the stability analysis of A5 = (1, 0, 1), in Figure 7a, when the incremental cost and free-rider income of the general contractor for green construction decrease, the general contractor’s trend toward non-green construction strategy will decrease. However, when the general contractor starts to trend toward a green construction strategy, because the developer’s parameters have not changed, it will ignore the government’s higher supervision. At this time, the benefits brought by the income and free-rider effect can compensate for the government’s higher punishment; thus, the ultimate stability point is (non-green construction, green construction, active supervision). To bring the equilibrium closer to the optimal strategy goal and reduce the incremental cost and free-rider revenue of green construction by the general contractor and the developer simultaneously (as shown in Figure 7b), at first, because the free-rider revenue is relatively high, one party of the developer and the general contractor will choose the non-green construction strategy in the game process; later, in the green construction system, one of the developers and the general contractor will first choose the green construction strategy. Due to the decrease in incremental cost and free-rider revenue and the higher government supervision, the developer will take the lead in the trend of the green construction strategy. Although the incremental cost and free-rider income of the general contractor are reduced, the developer still has the tendency to choose non-green construction due to the trend of green construction. At this time, although the government’s supervision of the general contractor is relatively low, the government’s probability of choosing active supervision will increase due to the general contractor’s tendency to choose non-green construction. However, for the general contractor, due to the increase in the government’s supervision probability, it will also have the trend of green construction. As both developers and general contractors have the trend of green construction, the government will tend to engage in negative supervision. Therefore, there is no stable and balanced state at this stage, and there is no ESS. At this time, the incremental cost and free-rider income of developers and general contractors are relatively low. At this point, they will choose the green construction strategy no matter what strategy the other party chooses. Therefore, the government will choose the passive supervision strategy. The simulation results in Figure 8 show that ESS changed from the original (green construction, non-green construction, active supervision) to green construction (green construction, passive supervision), thus realizing the optimization of the game system.

#### 5.1.3. Numerical Simulation Analysis of A6 (0, 1, 1)

According to the asymptotic stability requirement of A6 = (0, 1, 1), the green construction system is simulated in combination with the actual situation, where *x* = 0.5, *θ* = 0.4, *I*_1_ = 10, *I*_2_ = 8, *C*_1_ = 5, *C*_2_ = 4, *E*_1_ = 3, *E*_2_ = 3, *E*_3_ = 1.8, *E*_4_ = 1.3, *D*_1_ = 2.5, *D*_2_ = 2.4, *S*_1_ = 0.01, *S*_2_ = 0.01, *F*_1_ = 0.65 and *F*_2_ = 1.3. The results obtained by the simulation software are shown in Figure 9.

The initial dynamic evolution trend is stable at the system equilibrium point A6 = (0, 1, 1) (as shown in Figure 9), indicating (non-green construction, green construction, active supervision) as the ESS. According to the above analysis, when the government continues to increase its supervision, the first step is to increase the government’s supervision over developers, and the second step is to increase the government’s supervision over general contractors. According to the stability analysis of A6 = (0, 1, 1), it can be seen from Figure 10a that, when the government’s penalty parameter *F*_2_ is added to the general contractor, the general contractor will be more inclined to choose the green construction strategy. Although the developer will slow the speed of the non-green construction strategy, it will still be stable in the non-green construction strategy in the end. Simulation results show that the ESS is still (non-green construction, green construction, active supervision). As seen from Figure 10b, when the government’s penalty parameter *F*_1_ for developers is increased, with the increase in the parameter *F*_1_, similar to the above analysis, in the first stage, due to the existence of the free-rider effect, one party will choose the green construction strategy, and the other party will choose the non-green construction strategy. In the second stage, when the free-rider effect of the general contractor is less than the government penalty, the general contractor will choose the green construction strategy no matter how the developer chooses the strategy. However, due to the free-rider benefit of developers, when the general contractor chooses the green construction strategy, it will choose the non-green construction strategy. At this time, the green construction system ESS will change from (non-green construction, green construction, and active supervision) to (green construction, non-green construction, and active supervision). Therefore, the optimization of the system cannot be realized when the government continues to increase its supervision on this basis.

However, combined with the stability analysis of A6 = (0, 1, 1), in Figure 11a, when the excess revenue parameters *E*_1_ and *E*_3_ of the developers carrying out green construction are increased, due to the increase in the additional revenue brought by the developers carrying out green construction themselves, the tendency of the developers toward the non-green construction strategy will be reduced at first. Due to the free-rider effect, the developers and the general contractor will appear to have one party choosing the green construction strategy and the other party declining it. When the excess revenue parameters *E*_1_ and *E*_3_ of the developers for green construction continue to increase, the developers’ revenue is sufficient to make them give up the free-rider effect at this time. Therefore, no matter what strategy the general contractor chooses, the developers will choose the green construction strategy. As the general contractor’s parameters have not changed, the general contractor will ignore the government’s higher supervision intensity. At this time, the benefits brought by the revenue and free-rider effect can compensate for the government’s higher punishment. Therefore, the ultimate stability point is (green construction, non-green construction, active supervision). To bring the equilibrium closer to the optimal strategy goal, the excess revenue parameters *E*_1_, *E*_2_, *E*_3_, and *E*_4_ of the developers and the general contractor for green construction increase simultaneously (as shown in Figure 11b). First, because the additional revenue is relatively low, one of the developers and the general contractor will also choose the non-green construction strategy in the game process. Later, in the green construction system, one of the developers and the general contractor will first choose the green construction strategy. As the additional revenue rises, however, the government has a higher level of supervision over the general contractor. At this time, the general contractor will take the lead in the trend of the green construction strategy. On the basis of the comparison between the effect brought by the increase in additional revenue and the free-rider effect, and considering the government’s supervision, if the benefit brought by the free-rider effect is relatively large and the difference between the free-rider effect and the government’s punishment is greater than the effect brought by the additional revenue, the developer will choose the non-green construction strategy, and the simulation result ESS is (non-green construction, green construction, active supervision). If the benefit brought by the free-rider effect is relatively large but the difference between the free-rider effect and the government penalty is less than the effect brought by the excess benefit, the developers will start the game with the government. When the developers have the trend of green construction, the government will choose the negative supervision strategy, and the evolution path of the developers will change from the green construction direction to the non-green construction direction according to the change in the government strategy; thus, there is no stable equilibrium point in this state. With the increase, the excess revenue of the developer and the general contractor is relatively low. No matter what strategy the other party chooses, they will choose the green construction strategy. Therefore, the government will choose the passive supervision strategy. The simulation results in Figure 12 show that the ESS changed from the original (non-green construction, green construction, active supervision) to green construction (green construction, passive supervision) to optimize the game system.

### 5.2. The Impact of Government Regulation on Equilibrium Point Considering Consumers’ Green Preference

To further explore the impact of government regulation on the equilibrium point considering consumers’ green preferences, the regulation strategy was divided into four stages: low-quality supervision (*S*_1_ = 0.01, *S*_2_ = 0.01, *F*_1_ = 0.65, *F*_2_ = 0.65), lower-quality supervision (*S*_1_ = 0.4, *S*_2_ = 0.3, *F*_1_ = 1, *F*_2_ = 1.2), higher-quality supervision (*S*_1_ = 0.7, *S*_2_ = 0.5, *F*_1_ = 1.6, *F*_2_ = 1.4), and higher-quality supervision (*S*_1_ = 1, *S*_2_ = 0.8, *F*_1_ = 2, *F*_2_ = 1.7). Combined with the actual situation, other parameter values remained unchanged, the green construction system was simulated, and the results obtained by the simulation software are shown in Figure 13.

Under the condition that consumers’ green preference is less than 0.5, the stable equilibrium point of the green construction system changes from (0, 0, 1) to (1, 0, 1) or (0, 1, 1) with the improvement of supervision, indicating that, when consumers’ green preference is relatively low, the government’s active supervision can effectively mobilize the enthusiasm of developers and general contractors for green construction and promote the optimization of the green construction system.

When the consumer’s green preference is greater than 0.5 and less than 0.9, the stable equilibrium point of the green construction system changes from “available” to “non-available”, i.e., from (1, 0, 1) or (0, 1, 1) to “no equilibrium point”, indicating that when the consumer’s green preference is relatively high, one side of the developer or general contractor will “speculate”. When the government has a positive regulatory trend, it will evolve toward green construction. When the government has a negative regulatory trend, it will evolve toward non-green construction. At this time, there is no stable equilibrium point. Under the condition of lower supervision quality, the instability increases with the increase in consumers’ green preferences. Under higher quality supervision, the instability shows a “U”-shaped trend with the increase in consumers’ green preferences. Under high-quality supervision, the instability shows an “N”-shaped trend with the increase in consumers’ green preferences.

When the consumer’s green preference is greater than 0.9, the developer and the general contractor will have the highest revenue when they choose to carry out green construction. Therefore, the developer or the general contractor will choose to carry out green construction regardless of the strategy adopted by the government and the general contractor or the developer. Since both the developer and the general contractor will choose green construction, the government will choose negative supervision. At this time, the stability point of the green construction system is (green construction, green construction, negative supervision), and the system will reach its optimal state.

## 6. Conclusions and Suggestions

On the basis of the characteristics of bounded rationality and information asymmetry, this paper applied evolutionary game theory to the research of green construction supervision, established a three-party evolutionary game model with developers, general contractors, and the government as the main body, and analyzed the stability of the evolution strategy of the single-party system by copying the dynamic equations obtained in the model. The validity of the evolutionary game was verified. According to the simulation results, (1) A4 = (0, 0, 1), A5 = (1, 0, 1), A6 = (0, 1, 1), and A7 = (1, 1, 0) are stable equilibrium points, among which A7 = (1, 1, 0) is an ideal stable equilibrium point. (2) If consumer preference is relatively low, the government can strengthen the supervision, which can effectively mobilize the enthusiasm of the enterprises to participate so that one of the developers or the general contractor can put into the green construction system, and then the enterprises of both parties can actively carry out technological innovation and equipment improvement to improve the consumer’s green preference so that both parties can put into the green construction system. When consumers’ preferences are high, the government should advocate green consumption to promote the improvement of consumers’ green preferences so that developers and general contractors can invest in the green construction system. For the government, when both developers and general contractors carry out green construction, the sum of the supervision cost when the government invests in supervision and the subsidy to the enterprise is greater than the fine received from the non-green construction enterprise, and the government will choose negative supervision. (3) When consumer preferences change, ESS changes accordingly. Consumer preference plays a positive role in the establishment and development of green construction, but a certain increase in consumer preference will lead to free-rider behavior. (4) After adjusting the parameters to optimize the game system, A4 = (0, 0, 1), A5 = (1, 0, 1), and A6 = (0, 1, 1) can all reach the ideal state.

Comprehensive research showed that the optimal ESS is green construction and negative supervision, which is the optimal strategic goal. The key factors affecting the game system are consumers’ green preferences, enterprises’ incremental costs, free-rider income, excess income, and government supervision. Combined with the above research results, to improve the enthusiasm of enterprises to participate in green construction and promote the establishment and development of green construction systems, some suggestions are put forward.

First, the key to building a green construction system for enterprises is to cooperate in green production and operation. For this reason, enterprises should focus on long-term interests, establish a green consensus, strengthen contract design and system construction, and promote green coordination and cooperation among enterprises. The excess returns and incremental costs of enterprises in the green construction system, the opportunism caused by information asymmetry and the demonstration and leading role of enterprises actively carrying out green construction all have a significant impact on the development of the green construction system. Therefore, enterprises in the green construction system should learn from and introduce advanced green construction systems and management methods, formulate reasonable benefit distribution and punishment mechanisms, strengthen the construction of information sharing platforms among enterprises in the construction supply chain, promote the transparency of information in the production, supply and sales systems of green products, and restrict the opportunistic behavior of enterprises, as well as increase the support of green construction enterprises for the capital, technology and management innovation of green construction of other enterprises in the green construction system, promote the research and development and innovation of green construction technology, and promote the cooperation of enterprises in the green construction system to realize the greening of production, operation, and management.

Second, consumers’ preferences for green products should be improved. On the one hand, in the green construction system of the construction supply chain, enterprises should make full use of various media, intensify the publicity of green buildings, expand the consumption market of green buildings, increase consumers’ attention to green buildings, and urge consumers to consume them. At the same time, they can also enhance consumers’ awareness of green buildings and their loyalty and satisfaction with green building products. On the other hand, the government should strengthen the promotion of the necessity of ecological environment protection to consumers, improve their green preferences, and provide them with certain green consumption subsidies to further guide them to green consumption.

Third, the government’s reward and punishment mechanism should be improved. On the one hand, the government should give reasonable rewards and punishments to enterprises based on the green consumption demand of consumers, further improve and perfect the government’s punishment mechanism, establish tax incentives and subsidy mechanisms, and implement the enterprise green construction responsibility system. On the other hand, to strengthen the supervision and management of green construction, only under the promotion of green technology, improvement of the relevant laws, regulations, and standardization system, and further strengthening of the policy guidance of green construction can the participating competitors be at the same base point, compete for the same goal at the same cost, solve the restriction of free-rider benefits in the promotion process, help enterprises continue to promote green construction, and realize the establishment and sustainable development of green construction systems.

## Figures and Tables

**Figure 1 ijerph-19-16743-f001:**
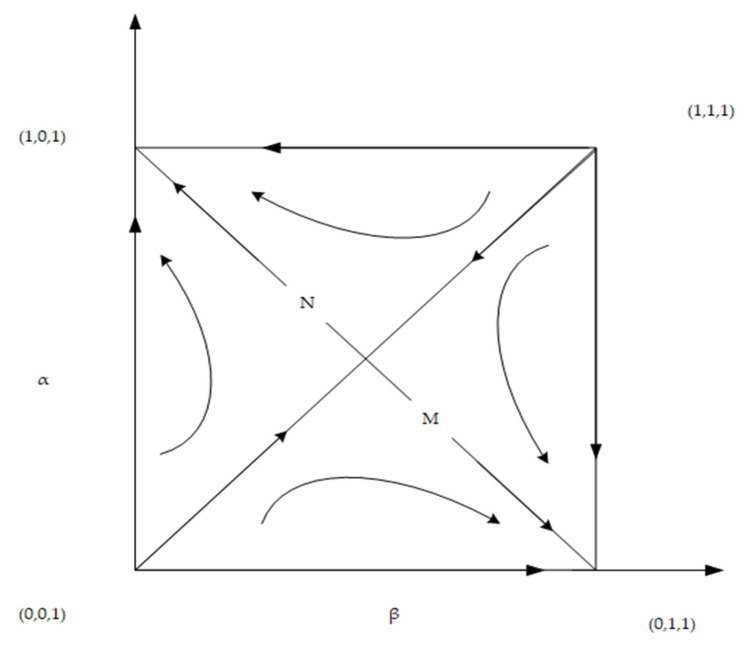
Evolutionary phase diagram.

**Figure 2 ijerph-19-16743-f002:**
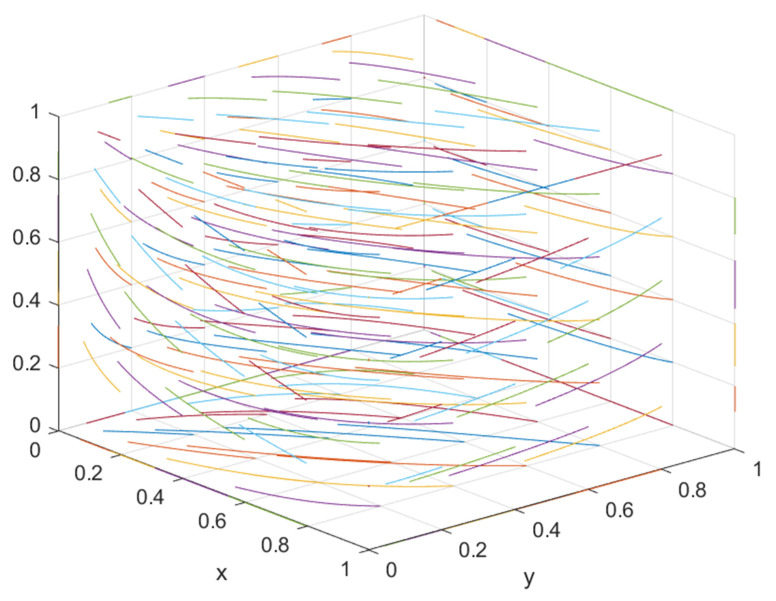
A4 = (0, 0, 1) evolution result.

**Figure 3 ijerph-19-16743-f003:**
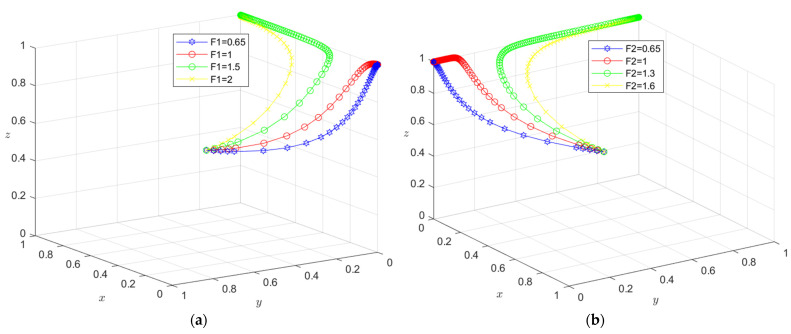
(**a**) Evolution path of the developer, general contractor and government under different government punishments *F*_1_; (**b**) evolution path of the developer, general contractor, and government under different government punishments *F*_2_.

**Figure 4 ijerph-19-16743-f004:**
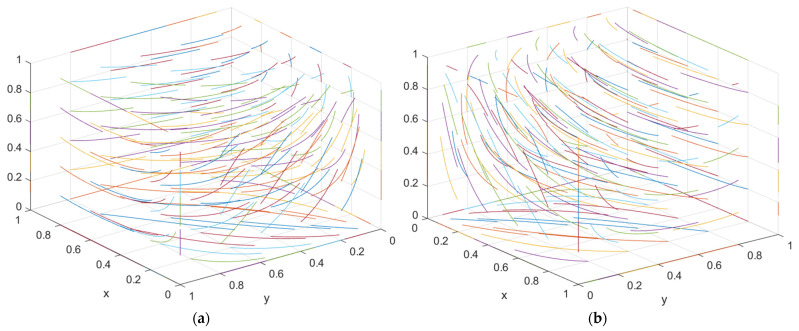
(**a**) Evolution of developers, general contractors, and governments. Parameters: *x* = 0.5, *θ* = 0.4, *I*_1_ = 10, *I*_2_ = 8, *C*_1_ = 5, *C*_2_ = 4, *E*_1_ = 3, *E*_2_ = 3, *E*_3_ = 1.8, *E*_4_ = 1.3, *D*_1_ = 2.5, *D*_2_ = 2.4, *S*_1_ = 0.01, *S*_2_ = 0.01, *F*_1_ = 1.5 and *F*_2_ = 0.65. (**b**) Evolution of developers, general contractors, and governments. Parameters: *x* = 0.5, *θ* = 0.4, *I*_1_ = 10, *I*_2_ = 8, *C*_1_ = 5, *C*_2_ = 4, *E*_1_ = 3, *E*_2_ = 3, *E*_3_ = 1.8, *E*_4_ = 1.3, *D*_1_ = 2.5, *D*_2_ = 2.4, *S*_1_ = 0.01, *S*_2_ = 0.01, *F*_1_ = 0.65 and *F*_2_ = 1.3.

**Figure 5 ijerph-19-16743-f005:**
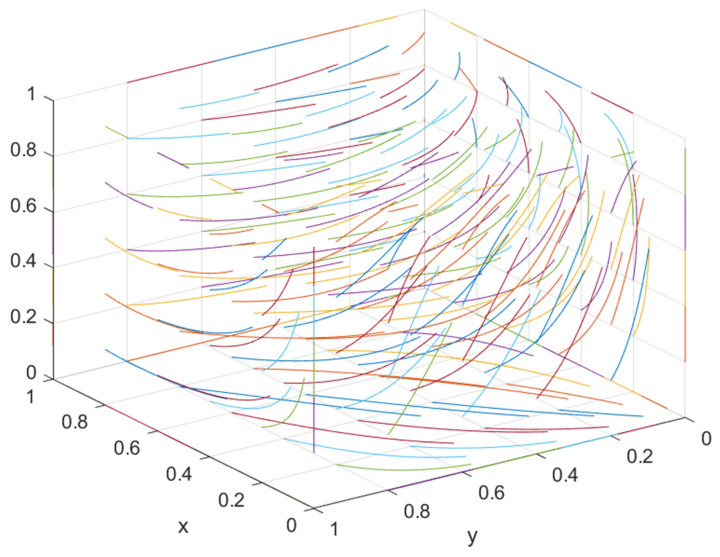
A5 = (1, 0, 1) evolution result.

**Figure 6 ijerph-19-16743-f006:**
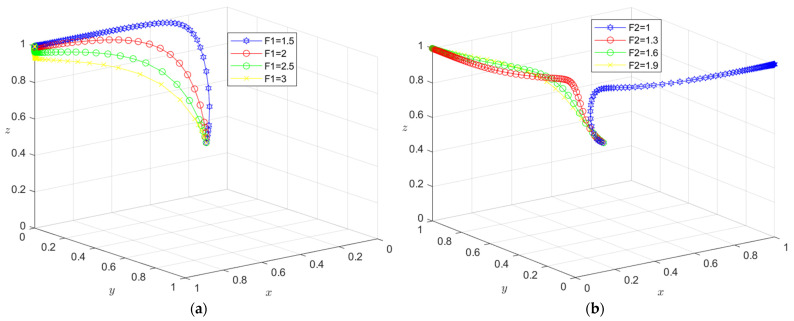
(**a**) Evolution path of the developer, general contractor, and government under different government punishments *F*_1_. (**b**) Evolution path of the developer, general contractor, and government under different government punishments *F*_2_.

**Figure 7 ijerph-19-16743-f007:**
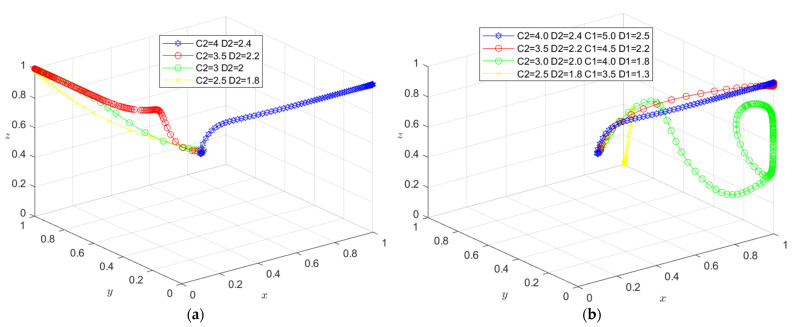
(**a**) Evolution path of the developer, general contractor, and government under different government punishments *C*_2_, *D*_2_. (**b**) Evolution path of the developer, general contractor, and government under different government punishments *C*_1_, *C*_2_, *D*_1_, *D*_2_.

**Figure 8 ijerph-19-16743-f008:**
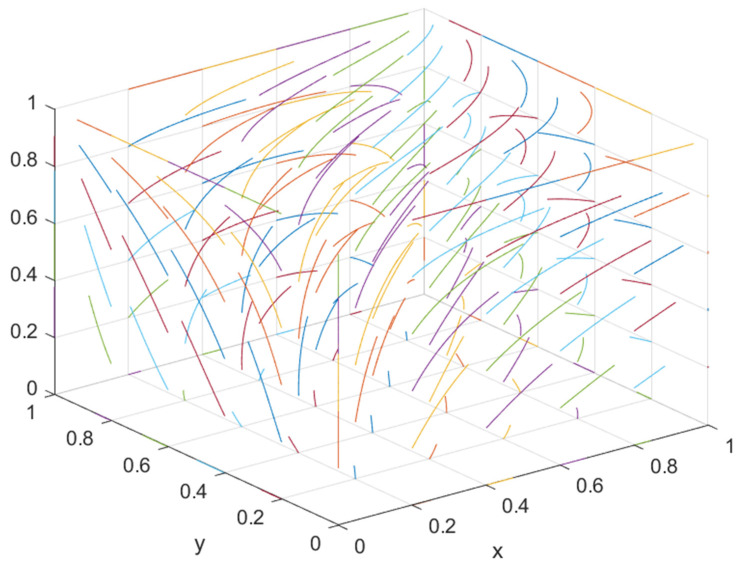
Evolution of developers, general contractors and governments. Parameters: *x* = 0.5, *θ* = 0.4, *I*_1_ = 10, *I*_2_ = 8, *C*_1_ = 3.5, *C*_2_ = 2.5, *E*_1_ = 3, *E*_2_ = 3, *E*_3_ = 1.8, *E*_4_ = 1.3, *D*_1_ = 1.3, *D*_2_ = 1.8, *S*_1_ = 0.01, *S*_2_ = 0.01, *F*_1_ = 1.5 and *F*_2_ = 0.65.

**Figure 9 ijerph-19-16743-f009:**
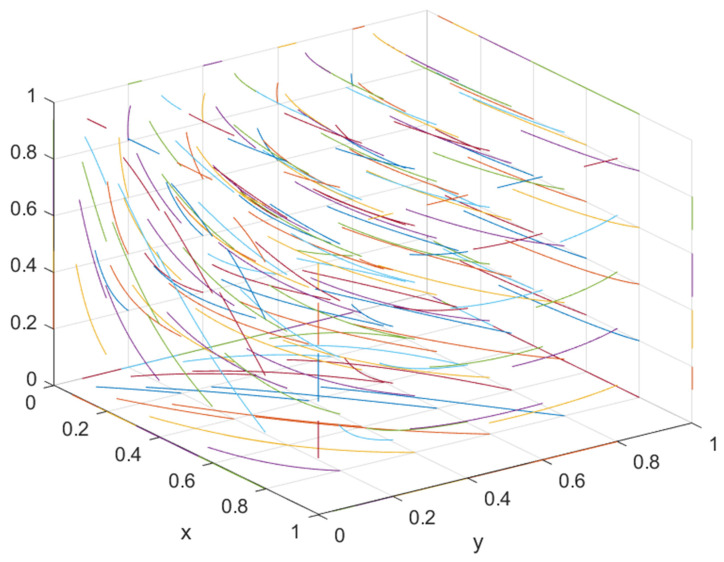
A6 = (0, 1, 1) evolution result.

**Figure 10 ijerph-19-16743-f010:**
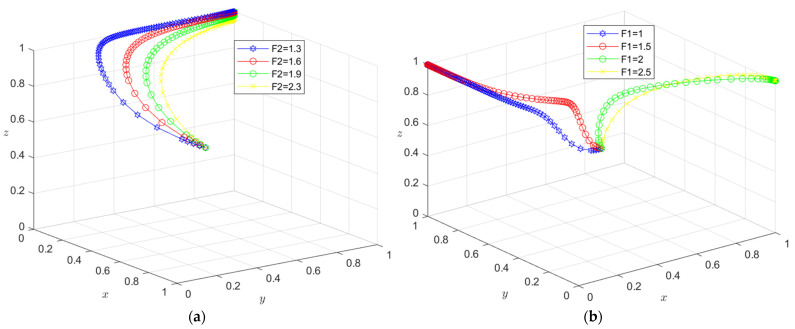
(**a**) Evolution path of the developer, general contractor, and government under different government punishments *F*_2_. (**b**) Evolution path of the developer, general contractor, and government under different government punishments *F*_1_.

**Figure 11 ijerph-19-16743-f011:**
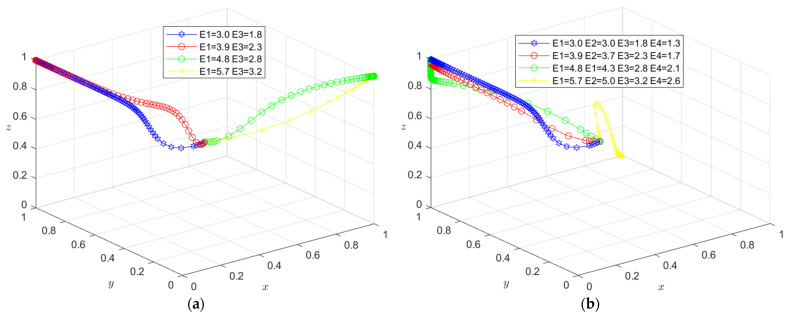
(**a**) Evolution path of the developer, general contractor and government under different government punishments *E*_1_, *E*_3_. (**b**) Evolution path of the developer, general contractor, and government under different government punishments *E*_1_, *E*_2_, *E*_3_, *E*_4_.

**Figure 12 ijerph-19-16743-f012:**
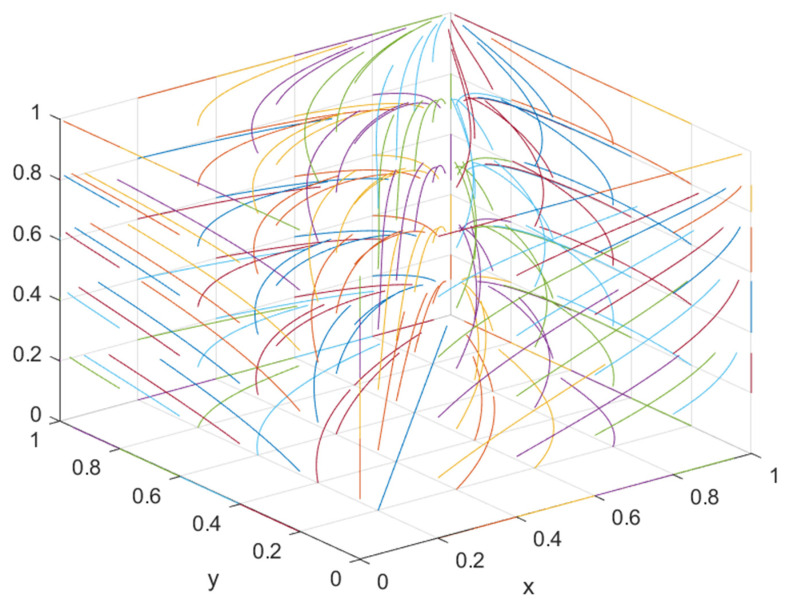
Evolution of developers, general contractors and governments. Parameters: *x* = 0.5, *θ* = 0.4, *I*_1_ = 10, *I*_2_ = 8, *C*_1_ = 3.5, *C*_2_ = 2.5, *E*_1_ = 5.7, *E*_2_ = 5.0, *E*_3_ = 3.2, *E*_4_ = 2.6, *D*_1_ = 1.3, *D*_2_ = 1.8, *S*_1_ = 0.01, *S*_2_ = 0.01, *F*_1_ = 0.65 and *F*_2_ = 1.3.

**Figure 13 ijerph-19-16743-f013:**
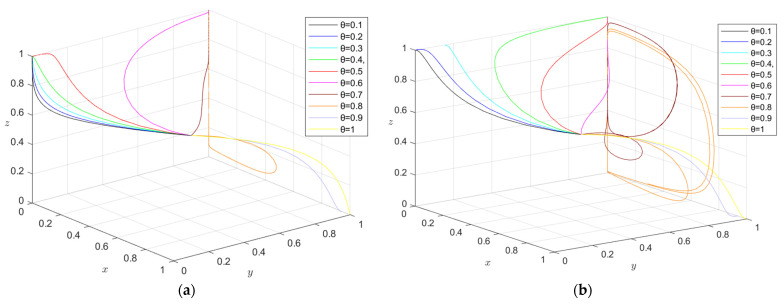
(**a**) Evolution path of developer, general contractor, and government under low supervision quality; (**b**) evolution path of developers, general contractors, and government under lower regulatory quality; (**c**) evolution path of developers, general contractors, and government under higher supervision quality; (**d**) evolution path of developer, general contractor, and government under high supervision quality.

**Table 1 ijerph-19-16743-t001:** Main body parameters.

Main Body	Parameters	Explanatory Notes
Developer	*I* _1_	Revenue from normal construction
	*C* _1_	Incremental cost of green construction
	*E* _1_	Excess revenue from green construction by both parties
	*E* _3_	Excess revenue from green construction only by oneself
	*D* _1_	Spillover income
	*S* _1_	government subsidy
	*F* _1_	Government punishment
	*x*	Contribution to green construction
General contractor	*I* _2_	Revenue from normal construction
	*C* _2_	Incremental cost of green construction
	*E* _2_	Excess revenue from green construction by both parties
	*E* _4_	Excess revenue from green construction only by oneself
	*D* _2_	Spillover income
	*S* _2_	government subsidy
	*F* _2_	Government punishment
	1 − *x*	Contribution to green construction
Government	*C* _3_	Regulatory costs
	*C* _4_	Environmental treatment costs incurred by developers due to non-green construction
	*C* _5_	Environmental governance costs incurred by general contractors for non-green construction
Consumer	*θ*	Consumer preference for green building products

**Table 2 ijerph-19-16743-t002:** Matrix of benefits for game subjects.

	Active Supervision (γ)		Negative Supervision (1−γ)	
	Green Construction (β)	Non-Green Construction (1−β)	Green Construction (β)	Non-Green Construction (1−β)
Green construction (α)	$(1+\theta )I_{1}+E_{1}-C_{1}+S_{1}$	$I_{1}+x\theta I_{1}+E_{3}-C_{1}+S_{1}$	$(1+\theta )I_{1}+E_{1}-C_{1}$	$I_{1}+x\theta I_{1}+E_{3}-C_{1}$
	$-C_{3}-S_{1}-S_{2}$	$-C_{3}-S_{1}+F_{2}-C_{5}$	0	$-C_{5}$
	$(1+\theta )I_{2}+E_{2}-C_{2}+S_{2}$	$I_{1}+x\theta I_{1}+D_{2}-F_{2}$	$(1+\theta )I_{2}+E_{2}-C_{2}$	$I_{1}+x\theta I_{1}+D_{2}$
Non-green construction (1−α)	$I_{1}+\left(1-x\right)\theta I_{1}+D_{1}-F_{1}$	$I_{1}$ − $F_{1}$	$I_{1}+\left(1-x\right)\theta I_{1}+D_{1}$	$I_{1}$
	$-C_{3}-C_{4}+F_{1}-S_{2}$	$-C_{3}-C_{4}-C_{5}+F_{1}+F_{2}$	$-C_{4}$	$-C_{4}-C_{5}$
	$I_{2}+\left(1-x\right)\theta I_{2}+E_{4}-C_{2}+S_{2}$	$I_{2}$ − $F_{2}$	$I_{2}+\left(1-x\right)\theta I_{2}+E_{4}-C_{2}$	$I_{2}$

**Table 3 ijerph-19-16743-t003:** Equilibria points and characteristic values.

	$\lambda_{1}$	$\lambda_{2}$	$\lambda_{3}$
(0, 0, 0)	$x\theta I_{1}+E_{3}-C_{1}$	$\left(1-x\right)\theta I_{2}+E_{4}-C_{2}$	$-C_{3}+F_{1}+F_{2}$
(1, 0, 0)	$-x\theta I_{1}-E_{3}+C_{1}$	$E_{2}-D_{2}+\left(1-x\right)\theta I_{2}-C_{2}$	$-S_{1}-C_{3}+F_{2}$
(0, 1, 0)	$x\theta I_{1}-C_{1}+E_{1}-D_{1}$	$-\left(1-x\right)\theta I_{2}-E_{4}+C_{2}$	$-S_{2}-C_{3}+F_{1}$
(0, 0, 1)	$\left(S_{1}+F_{1}\right)+x\theta I_{1}+E_{3}-C_{1}$	$\left(S_{2}+F_{2}\right)+\left(1-x\right)\theta I_{2}+E_{4}-C_{2}$	$C_{3}-F_{1}-F_{2}$
(1, 0, 1)	$-\left[\left(S_{1}+F_{1}\right)+x\theta I_{1}+E_{3}-C_{1}\right]$	$\left(S_{2}+F_{2}\right)+E_{2}-D_{2}+\left(1-x\right)\theta I_{2}-C_{2}$	$S_{1}+C_{3}-F_{2}$
(0, 1, 1)	$S_{1}+F_{1}+x\theta I_{1}-C_{1}+E_{1}-D_{1}$	$-\left[\left(S_{2}+F_{2}\right)+\left(1-x\right)\theta I_{2}+E_{4}-C_{2}\right]$	$S_{2}+C_{3}-F_{1}$
(1, 1, 0)	$-\left[x\theta I_{1}-C_{1}+E_{1}-D_{1}\right]$	$-[E_{2}-D_{2}+\left(1-x\right)\theta I_{2}-C_{2}]$	$-S_{1}-S_{2}-C_{3}$

**Table 4 ijerph-19-16743-t004:** Correlation Analysis of System Parameters.

Parametric Variation	*S_M_* Change	Evolutionary Direction
*x* ↓	*S_M_* ↑	(0, 1, 1)
*I*_1_ ↓	*S_M_* ↑	(0, 1, 1)
*I*_2_ ↑	*S_M_* ↑	(0, 1, 1)
*C*_1_ ↑	*S_M_* ↑	(0, 1, 1)
*C*_2_ ↓	*S_M_* ↑	(0, 1, 1)
*E*_1_ ↓	*S_M_* ↑	(0, 1, 1)
*E*_2_ ↑	*S_M_* ↑	(0, 1, 1)
*E*_3_ ↓	*S_M_* ↑	(0, 1, 1)
*E*_4_ ↑	*S_M_* ↑	(0, 1, 1)
*D*_1_ ↑	*S_M_* ↑	(0, 1, 1)
*D*_2_ ↓	*S_M_* ↑	(0, 1, 1)
*S*_1_ ↓	*S_M_* ↑	(0, 1, 1)
*S*_2_ ↑	*S_M_* ↑	(0, 1, 1)
*F*_1_ ↓	*S_M_* ↑	(0, 1, 1)
*F*_2_ ↑	*S_M_* ↑	(0, 1, 1)

## Data Availability

Not applicable.

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
