# Peer review of "Evolutionary Game Research on Green Construction Considering Consumers’ Preference under Government Supervision"

_ijerph, 2022, doi:10.3390/ijerph192416743_

Round 1

Reviewer 1 Report

1\Line 20, in the abstract, abbreviation ESS should be firstly introduced following evolutionary stability strategies in line 17.

2\Green building has some similarities or characteristics with construction industrialization, I recommend to refer to the driving factors of Construction Industrialization Development in China discussed in the following article. Xiahou, X.; Yuan, J.; Liu, Y.; Tang, Y.; Li, Q. Exploring the Driving Factors of Construction Industrialization Development in China. Int. J. Environ. Res. Public Health 2018, 15, 442. https://doi.org/10.3390/ijerph15030442

3\ In “3 model” development. Why the authors only select the general contractor, there may be other stakeholders, like ordinary constructers or sub constructers. And why the authors select government, general contract, and developers?

4\ too much equation in section 4, it is not quite necessary. Or could be submitted as supplicatory materials.

5\ An in-depth discussion should be made, especially the findings with previous research in the literature results.

Reviewer 2 Report

First, I would like to point out that I appreciate the work done by the authors, and it is not my intention to depreciate the job. And sorry for that, my opinion is very critical. The article submitted for review does not fall within the scope of the journal's interests. The only common denominator between the article's content and the journal's area of interest are the words "ecological" and "green", abused by the authors. However, the article's content concerns the possibility of using game theory in sustainable development in the construction industry. In the presented form, the paper is not suitable for publication.

NOTES:

In the Introduction (lines 29-31), the authors wrote, "This study provides theoretical support and practical guidance for the establishment and development of green construction and further promotes the healthy and sustainable development of green and low-carbon construction supply chains."

A manager is a person who is responsible for making quick and accurate decisions based on synthetic information, usually containing several possible variants of solving the problem. It is vain to look for such solutions in the article's content or the Conclusions and Suggestions chapter itself. However, analysing the paper's content requires highly specialized knowledge in mathematics, economics, game theory and aspects related to government supervision over the construction industry.

The frequent use of the phrase supply chain (e.g. keywords - line 33) is also an incomprehensible trick.

Each supply chain consists of three main stages:

1. Procurement - concerns the method, place and time of obtaining and delivering raw materials for the production of products.

2. Production - that is the transformation of raw materials into finished products.

3. Distribution - all activities enable the delivery of products to the destination.

Only a parallel analysis of these three stages can lead to conclusions regarding the entire supply chain. I didn't find it in the article. Please indicate the fragments in which the authors made such an analysis. If this is not possible, it is necessary to re-analyze which keywords should be included in the article description.

It is also necessary to supplement the References with the latest achievements in the described area.

General note:

The article's narrative is conducted in a complicated way and requires going backwards to recall the individual designations of the model. This causes severe fatigue and weariness for the reader, who is not an expert in this area.

Reviewer 3 Report

The article concerns a particularly important environmental and social issue of green construction. The subject of the article is an analysis of the relationship between developers, general contractors, and the government. A crucial factor considered in the relations of the game participants are green preferences. In the study of these relationships, the achievements of evolutionary game theory were used. In the empirical layer, simulation models and the MATLAB program were used.

Remarks:

The abstract should be more structured (purpose, study design/methodology/approach, findings, originality/value, research limitations/implications, practical implications).

The acronym ESS (Evolutionarily Stable Strategy) used in the abstract is developed further in the paper (as “evolutionary stability strategies”).

The acronym ESS (Evolutionarily Stable Strategy) was introduced by John Maynard Smith in 1972 in the chapter “Game Theory and the Evolution of Fighting”.

There are no references to important works of co-authors of the theory of evolutionary games, e.g.:

Simon, H.A. Rationality as process and as product of thought. Am. Econ. Rev. 1978, 68, 1-16. (Nobel Prize)

Smith, M. J. (1982). Evolution and the Theory of Games, Cambridge: Cambridge University Press.

As well as contemporary publications, e.g.:

Weibull, J. Evolutionary Game Theory; MIT Press: Cambridge, MA, USA, 1995.

Samuelson, L. Introduction to the evolution of preferences. J. Econ. Theory 2001, 97, 225-230. 

Newton, J. Evolutionary Game Theory: A Renaissance. Games 2018, 9, 31. https://doi.org/10.3390/g9020031. 

Round 2

Reviewer 2 Report

Thank you very much for the work done. I am impressed with the changes made based on the suggestions I sent. However, I still maintain the opinion that the article's narrative is conducted in a complicated way and requires going back to remember the individual designations of the model. It seems, however, that this may only be my subjective impression and should not be an argument for rejecting the article. I propose to accept the material for publication.